# INTERPRETABLE OUT-OF-DISTRIBUTION DETECTION USING PATTERN IDENTIFICATION

## ABSTRACT

Out-of-distribution (OoD) detection for data-based programs is a goal of paramount importance. Common approaches in the literature tend to train detectors requiring inside-of-distribution (in-distribution, or IoD) and OoD validation samples, and/or implement confidence metrics that are often abstract and therefore difficult to interpret. In this work, we propose to use existing work from the field of explainable AI, namely the PARTICUL pattern identification algorithm, in order to build more interpretable and robust OoD detectors for visual classifiers. Crucially, this approach does not require to retrain the classifier and is tuned directly to the IoD dataset, making it applicable to domains where OoD does not have a clear definition. Moreover, pattern identification allows us to provide images from the IoD dataset as reference points to better explain the confidence scores. We demonstrates that the detection capabilities of this approach are on par with existing methods through an extensive benchmark across four datasets and two definitions of OoD. In particular, we introduce a new benchmark based on perturbations of the IoD dataset which provides a known and quantifiable evaluation of the discrepancy between the IoD and OoD datasets that serves as a reference value for the comparison between various OoD detection methods. Our experiments show that the robustness of all metrics under test does not solely depend on the nature of the IoD dataset or the OoD definition, but also on the architecture of the classifier, which stresses the need for thorough experimentations for future work on OoD detection.

## 1 INTRODUCTION

A fundamental aspect of software safety is arguably the modelling of its expected operational domain through a formal or semi-formal specification, giving clear boundaries on when it is sensible to deploy the program, and when it is not. It is however difficult to define such boundaries for machine learning programs, especially for visual classifiers based on artificial neural networks (ANN), which are the subject of this paper. Indeed, such programs process high-dimensional data (images, videos) and are the result of a complex optimization procedure, but they do not embed clear failure modes that could get trigged in the case of an unknown distribution, with potentially dire consequences in critical applications.

Although it is difficult to characterize an operational distribution, one could still measure its dissimilarity with other distributions. In this context, Out-of-Distribution (OoD) detection - which aims to detect whether an input of an ANN is Inside-of-distribution (IoD) or outside of it - serves several purposes. It helps characterize the extent to which the ANN can operate outside a bounded dataset (which is important due to the incompleteness of the training set *w.r.t.* the operational domain). It also constitutes a surrogate measure of the generalization abilities of the ANN. Finally, OoD detection can help assess when an input is too far away from the operational domain, which prevents misuses of the program and increases its safety.

## 2 RELATED WORK AND CONTRIBUTION

**Out-of-distribution detection** The *maximum class probability* (MCP) obtained after softmax normalization of the classifier logits already constitutes a good baseline for OoD detection Hendrycks & Gimpel (2017). However, neural networks tend to be overconfident on their predictions Szegedy et al.

(2014); Lee et al. (2018a); Hein et al. (2019), even when they are wrong, which may result in false claims of confidence. Hence, the development of enhancements like temperature scaling Liang et al. (2017), ensemble learning Nguyen et al. (2020) or True-Class Probability learning Corbiere et al. (2021) (assuming such information is known). Other types of confidence measures have also been developed, with various operational settings. Note that in this work, we focus on methods that can apply to pre-trained classifiers. Therefore, we exclude methods such as Lee et al. (2018a); Hein et al. (2019); Hendrycks et al. (2019) - which integrate the learning of the confidence measure within the training objective of the model - or specific architectures from the field of Bayesian Deep-Learning that aim at capturing uncertainty by design Gal & Ghahramani (2016). While efficient, these approaches may prove costly or impractical in an industrial context where a lot of resources might have already been dedicated to obtain an accurate model for the task at hand.

Moreover, we make a distinction between methods that require a validation set composed of OoD samples for the calibration of hyper-parameters (OoD-specific), and methods that do not require such validation set and are therefore "OoD-agnostic" (Liu et al. (2020)).

*OoD-specific methods*. ODIN Liang et al. (2018) extends the effect of temperature scaling with the use of small adversarial perturbations, applied on the input sample, that aim at increasing the maximum softmax score. OoD detection is performed by measuring the gain in softmax score after calibrating the temperature value and the perturbation intensity on a validation set, so that perturbations lead to a greater margin for IoD data than for OoD data. Other approaches also attempt to capture the "normal" behaviour of the different layers of the classifier: Huang et al. (2021) states that the latent representations of OoD samples through a CNN classifier are clustered around a point called the *feature-space singularity* (FSS) which serves as a reference point for OoD detection. Similarly, Lee et al. (2018b) proposes a confidence score based on the Mahalanobis distance between a new sample and class conditional Gaussian distributions inferred from the training set. Both approaches operate upon multiple layers of the network and require the use of a set of OoD samples for calibrating the relative importance of each layer in the final confidence score.

*OoD-agnostic methods*. Liu et al. (2020) proposes a framework based on energy scores (using in practice the denominator of the softmax normalization function) which can be used either during inference on a pre-trained model or to fine-tuned the model for more discriminative properties. The Fractional Neuron Region Distance Hond et al. (2021) (FNRD) computes the range of activations for each neuron on the training set, then provides a score describing how many neurons are activated outside their boundaries for a given input. Finally, Attribution-Based Confidence Jha et al. (2019) (ABC) does not requires access to IoD data and equates the confidence of the network to its local stability by sampling the neighbourhood of a given input and measuring stability through attribution methods. Although this last method relies on fewer prerequisites, it is computationally expensive (*e.g.,* gradient computation) and may not be suited to runtime constraints.

As noted by Hendrycks et al. (2019), OoD-specific and OoD-agnostic methods are usually "*not directly comparable*" due to their different prerequisites. Therefore, in this work, we mainly compare OoD-agnostic methods, using FSSD Huang et al. (2021) only as a reference measure to illustrate possible detection gaps between OoD-agnostic and OoD-specific methods. We also exclude ABC Jha et al. (2019) from our experiments due to its computational cost.

It is also important to note that all the methods presented above are evaluated on different datasets and definitions of OoD (*e.g.,* different datasets, distribution shifts), which only gives a partial picture on their robustness Tajwar et al. (2021). Although works such as Open-OoD Yang et al. (2022) - which aims at standardizing the evaluation of OoD detection, anomaly detection and open-set recognition into a unified benchmark - are invaluable for the community, most datasets commonly in use (MNIST Deng (2012), CIFAR-10/100 Krizhevsky (2009)) contain images with low resolution that may not reflect the detection capabilities of the methods under test in more realistic operational settings. Moreover, when evaluating the ability of an OoD detection method to discriminate between IoD and OoD datasets, it is often difficult to properly quantify the discrepancy between these two datasets, *independently* from the method under test, and therefore to exhibit a "ground truth" value of what this margin should be. Therefore, in this paper we propose a new type of OoD benchmark based on perturbations of the IoD dataset which aims at measuring the correlation between the OoD detection score of a given method on the perturbed dataset (OoD) and the intensity of the perturbation, under the hypothesis that the intensity of the perturbation can serve as a ground-truth measure of the discrepancy between the IoD and the OoD dataset.

**Part detection** Many object recognition methods have focused on part detection, in supervised (using annotations Farhadi et al. (2009); Han et al. (2018); Zhao et al. (2019)), weakly-supervised (using class labels Li et al. (2020); Peng et al. (2018)) or unsupervised settings (Zheng et al. (2017); Han et al. (2022); Xu-Darme et al. (2022)), primarily with the goal of improving accuracy on hard classification tasks such as fine-grained recognition Welinder et al. (2010); Yang et al. (2015). To our knowledge, the PARTICUL algorithm described in Xu-Darme et al. (2022) is the only method including a confidence measure associated with the detected parts (in the paper, it is used to infer the visibility of a given part). PARTICUL aims at identifying recurring patterns in the latent representation of a set of images processed through a CNN, in an unsupervised manner. It is however restricted to fine-grained recognition datasets where all images belong to the same macro-category.

**Our contributions** In this paper, we aim to investigate the following research tracks: 1) How well can OoD detection methods generalize across multiple definitions of OoD? 2) How to improve the interpretability of confidence measures in the context of visual classifiers? As an answer to those questions, we make the following contributions:

1. We show that recent work on pattern identification (Xu-Darme et al. (2022)) can be adapted to build an OoD detection measure that does not require any fine-tuning of the original classifier, and with performance on par with other OoD-agnostic methods. More importantly, pattern identification allows us to provide images from the IoD dataset as reference points to better explain the confidence scores;

2. We demonstrate the ability of our approach to consistently capture various notions of OoD over an extensive benchmark comprising four datasets, two architectures and several evaluation metrics. In particular, we introduce a new benchmark based on perturbations of the IoD dataset which provides a known and quantifiable evaluation of the discrepancy between the IoD and OoD datasets that serves as a reference value for the comparison between various OoD detection methods.

This paper is organized as follows: Sec. 3 formulates the problem and the two OoD modalities we will explore; Sec. 4 introduces our modifications to the PARTICUL algorithm with the goal of providing a more interpretable and robust confidence measure; Sec. 5 describes our experimental setup and results. Finally, Sec. 6 presents our conclusions and future work.

## 3    PROBLEM FORMULATION

Let $\mathcal{I}_s$ be the space of all RGB images of size $s$. Let $D_{iod} \subseteq \mathcal{I}_s$ be a dataset approximating the distribution $\mathcal{D}_{iod}$ of images belonging to $N$ categories. Let $M$ be a deep convolutional neural network (CNN) classifier trained on a subset of $D_{iod}$, $X_{train}$. For any image $x \in \mathcal{I}_s$, $M$ outputs a vector of logits $M(x) \in \mathbb{R}^N$, where the index of the highest value corresponds to the most probable category (or *class*) of $x$ - relatively to all other categories. Since any image is assigned a category regardless of its similarity to the training distribution, $M(x)$ is often normalized using a softmax function in order to indicate a level of confidence in the prediction. More generally, a confidence measure $C_M : \mathcal{I}_s \rightarrow [0, 1]$ is a function assigning a score to each image $x \in \mathcal{I}_s$ processed through $M$. We denote $C_M(\mathcal{D}_{iod})$ the distribution of values computed over $\mathcal{D}_{iod}$ using the confidence measure $C_M$. As indicated above, $C_M(\mathcal{D}_{iod})$ is approximated by $C_M(\mathcal{D}_{iod}) \approx C_M(D_{iod}) = \{C_M(x) \mid x \in D_{iod}\}$. Ideally, images belonging to the training distribution $\mathcal{D}_{iod}$ should have a higher confidence score than images outside $\mathcal{D}_{iod}$. We propose to take an empirical approach, which will only attempt to provide evidence that a confidence measure tends to return a greater score for images in $D_{iod}$ than images outside $D_{iod}$. However, it is not possible to obtain a dataset representative of $\mathcal{I}_s \setminus \mathcal{D}_{iod}$, *i.e.*, encompassing all possible OoD inputs. Therefore, in this work we use two weaker modalities of OoD detection that serve as benchmarks for assessing the various proposed confidence measures.

**Cross-dataset detection**    In this experiment, as in most related works, we evaluate the separability of the distributions $C_M(D_{iod})$ and $C_M(D_{ood})$, where $D_{ood}$ is a dataset different (not drawn from the same distribution) from $D_{iod}$. We use three complementary metrics that do not require determination of an acceptance threshold between IoD and OoD inputs, and that are suitable for comparing the different confidence measures proposed: the Area Under the ROC curve (AUROC); the Area Under the Precision-Recall curve (AUPR). As in Huang et al. (2021), we also compute the False Positive Rate when the true positive rate is 80% (FPR80).

**Sensitivity to dataset perturbation**   For this second experiment, we generate an OoD dataset $D_{ood}$ by applying a perturbation to all images of $D_{iod}$. More precisely, a perturbation $P_\lambda$ is a function which applies a transformation to an image $x \in \mathcal{I}_s$ with a variable magnitude $\lambda$ (*e.g.,* for a rotation, $\lambda$ might correspond to the angle of the rotation). Note that there exist no general value for the magnitude of the perturbation that would result in $D_{ood}$ to be OoD in all cases. In this paper, the maximum perturbation amplitude is chosen when the resulting image is qualitatively considered dissimilar to the base image. For a perturbation $P_\lambda$ applied over $D_{iod}$ , we define the average value of confidence measure $C_M$ as

$$\Gamma(C_M, D_{iod}, P, \lambda) = \frac{1}{|D_{iod}|} \sum_{x \in D_{iod}} C_M(P_\lambda(x)) \tag{1}$$

which is extended to a set of perturbation magnitudes $\Lambda = (\lambda_0, \ldots, \lambda_n)$ as $\Gamma(C_M, D_{iod}, P, \Lambda) = \big(\Gamma(C_M, D_{iod}, P, \lambda_0), \ldots, \Gamma(C_M, D_{iod}, P, \lambda_n)\big)$. Although it would again be possible to measure the separability of IoD and OoD confidence distributions, perturbations of small intensities would result in almost identical distributions. Instead, we evaluate the correlation between the intensity of the perturbation and the average confidence value of the perturbed dataset by computing the Spearman Rank correlation coefficient $r_s$ between $\Lambda$ and $\Gamma(C_M, D_{iod}, P, \Lambda)$. Indeed, $r_s\big(\Lambda, \Gamma(C_M, D_{iod}, P, \Lambda)\big) = 1$ (resp. $-1$) indicates that the average confidence measure increases (resp. decreases) monotonically with the value of $\lambda$, which indicates that the measure is at least correlated with an increase in the perturbation. The use of the Spearman Rank correlation score, rather than another qualitative metric such as the average value of the derivative $\frac{\delta \Gamma}{\delta \lambda}$ is motivated by the following reasons: 1) Averaging the derivative across multiple intensity values might hide the fact that there is no correlation between the measure and the perturbation; 2) A higher sensitivity of a given measure to a perturbation would not necessarily imply a better measure, especially when the different confidence measures under test might have different calibrations. Therefore, under the hypothesis that the discrepancy between $D_{iod}$ and $D_{ood}$ is correlated to the magnitude of the perturbation $\lambda$, this benchmark measures the consistency of the OoD method under test.

## 4   PROPOSED MODEL

In this section, we present two proposals for building a confidence measure based on the PARTICUL algorithm, as described in Xu-Darme et al. (2022). PARTICUL is intended to mine recurring patterns in the latent representation of a set of images processed through a CNN. In the context of this paper, patterns are learnt from the last convolutional layer of a classifier $M$ over the training set $D_{iod}$, in a plug-in fashion that does not require the classifier to be retrained. Let $F$ be the restriction of classifier $M$ up to its last convolutional layer, *i.e.,* $M = L \circ F$, where $L$ corresponds to the last pooling layer followed by one or several fully connected layers. $\forall x \in \mathcal{I}_s$, $F(x) \in \mathbb{R}^{H \times W \times D}$ is a convolutional map of $D$-dimensional vectors.

***Vanilla* PARTICUL**   The purpose of the PARTICUL algorithm is to learn $p$ distinct $1 \times 1 \times D$ convolutional kernels $K = [k^{(1)}, \ldots, k^{(p)}]$ (or *detectors*), such that $\forall x \in X_{train}$, each kernel $k^{(i)}$ strongly correlates with exactly one vector $F_{[h,w]}(x)$ in $F(x)$ (*Locality* constraint). After the training of said kernels, PARTICUL uses the function $H^{(i)}(x) = \max_{h,w} \big(F_{[h,w]}(x) * k^{(i)}\big)$ returning the maximum correlation score between kernel $k^{(i)}$ and the convolutional map $F(x)$. In Xu-Darme et al. (2022), the distribution $H^{(i)}(\mathcal{D}_{iod})$ is modeled as a random variable following a normal distribution $\mathcal{N}(\mu^{(i)}, \sigma^{(i)})$ and estimated (or *calibrated*) over $X_{train}$. Then, the corresponding confidence measure is computed using the cumulative distribution function (CDF) associated with $\mathcal{N}(\mu^{(i)}, \sigma^{(i)})$. However, such CDF does not have a closed-form solution, which can slow down operations on tensors. Therefore, in this work, we choose to model $H^{(i)}(\mathcal{D}_{iod})$ as a logistic distribution $\mathcal{L}(\mu^{(i)}, \sigma^{(i)})$, where $\mu^{(i)}$ and $\sigma^{(i)}$ are again estimated over $X_{train}$, and define the confidence associated with detector $i$ for image $x$ as:

$$C^{(i)}(x) = \frac{1}{1 + e^{-\frac{H^{(i)}(x) - \mu^{(i)}}{\sigma^{(i)}}}} \tag{2}$$

Finally, we average the confidence measures and define, $\forall x \in \mathcal{I}_s$, $C_M^{vP}(x) = \frac{1}{p} \sum_{i=1}^{p} C^{(i)}(x)$.

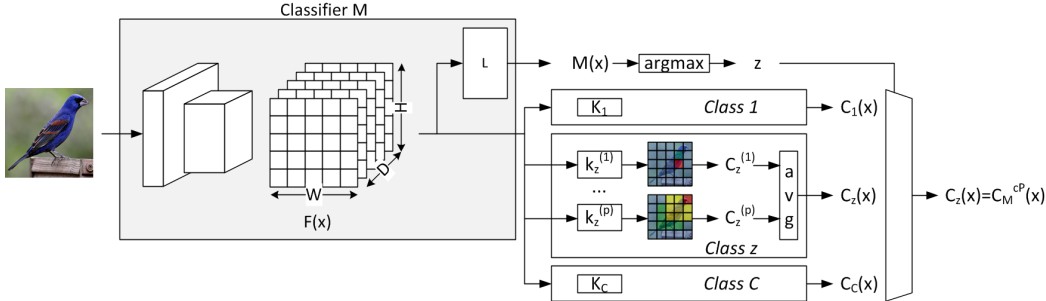

Figure 1: Class-based PARTICUL overview. When processing a new sample, the confidence measure averages the contribution of the detectors corresponding to the predicted class $z$ only.

PARTICUL is well-suited for datasets with low intra-set variance - *i.e.,* when images in $X_{train}$ are very similar - which is the case for fine-grained recognition datasets (Welinder et al. (2010); Yang et al. (2015)). However, for more heterogeneous datasets (Krizhevsky (2009); Li et al. (2022)), it becomes more difficult to find recurring patterns that are present across the entire training set. Moreover, in order to learn more consistent patterns, it might be beneficial to learn detectors that are specific to each of the $N$ classes of $\mathcal{D}_{iod}$.

***Class-based* PARTICUL**   Here, we introduce a variant of PARTICUL that uses the labels of the training set $X_{train}$ in order to learn $p$ detectors *per class*. During the training and calibration phases, for each pair of images and class label $(x, c) \in X_{train} \times \mathbb{N}$, only the detectors of class $c$ are modified. More precisely, let $K_c = [k_c^{(1)}, \dots k_c^{(p)}]$ be the set of kernel detectors for class $c$. As in Xu-Darme et al. (2022), $K_c$ is learned by minimizing the weighted combination of a locality function $\mathcal{L}_l$ and an unicity function $\mathcal{L}_u$, $\mathcal{L}(K_c) = \mathcal{L}_l(K_c) + \lambda \mathcal{L}_u(K_c)$, restricted to $X_{train,c} = \{x | (x, c) \in X_{train}\}$. Importantly, since patterns may be similar across different classes (*e.g.,* the wheels on a car or a bus), we do not treat images from other classes as negative samples when learning our class-based detectors. Therefore, although this approach may allow more specific detectors to be learnt than the *vanilla*-PARTICUL approach, it effectively restricts the number of training samples per detector to the images of a single class, which is potentially detrimental to the PARTICUL training process. The calibration of the detectors for class $c$ also uses a logistic distribution model, which is estimated over training samples from class $c$ only. Then, as shown in Fig. 1, for any image $x \in \mathcal{I}_s$, and with no access to the ground truth label of $x$, the confidence measure $C_M^{cP}(x)$ is obtained by averaging the confidence scores from the detectors associated with the predicted class $z = \text{argmax}\big(M(x)\big)$ of image $x$, *i.e.,* $C_M^{cP}(x) = \frac{1}{p} \sum_{i=1}^{p} C_z^{(i)}(x)$.

As a consequence, for an image $x \in \mathcal{D}_{iod}$, the quality of our class-based confidence measure is also dependent on the ability of the classifier to assign the correct class to $x$, *i.e.,* on the accuracy of $M$.

## 5   EXPERIMENTS

**Setup**   We performed our evaluation using a ResNet50 He et al. (2016) classifier trained on various datasets, with different image sizes: the Caltech-101 Li et al. (2022) dataset contains 9,144 images - from 101 different categories - that are resized to $224 \times 224$ during training (test accuracy 86.8%); the CUB-200 Welinder et al. (2010) dataset contains 11,788 images - from 200 different bird species - that are resized to $448 \times 448$ during training (test accuracy 83.6%); the Stanford Cars Yang et al. (2015) dataset contains 16,185 images - from 196 different car models - that are resized to $448 \times 448$ (test accuracy 90.5%). Note that additional experiments were conducted on the CIFAR-100 Krizhevsky (2009) dataset but are not detailed here (see the supplementary material) due to their similarity with the results obtained on Caltech-101. Both CUB-200 and Stanford Cars are fine-grained classification datasets and contain very homogeneous data from a single macro-category, while Caltech-101 contains more heterogeneous data from diverse categories. Each dataset is split between a training set $X_{train}$ (also used for PARTICUL calibration) and a test set $X_{test}$ (used for evaluation). We follow the dataset providers' proposed splits. Using the trained classifiers for feature

extraction, we implemented our PARTICUL detectors using the two approaches described above: for *vanilla* PARTICUL, we trained 4 or 6 global detectors, ignoring image labels and using the loss functions ($\mathcal{L}_l$ and $\mathcal{L}_u$) and learning parameters indicated in Xu-Darme et al. (2022); for *class-based* PARTICUL, we trained 4 or 6 detectors *per class*, using the image labels of the training set and the same loss functions/learning parameters, but on a per-class basis. Note that although this approach leads to a high number of detectors, the learning process remains computationally efficient since the weights of the classifier are not modified and only the detectors of the expected class are updated during the back-propagation phase. As a comparison, we also implemented a confidence measure $C_M^{fnrd}(x) = 1 - fNRD(x)$ based on Hond et al. (2021) (fNRD), as well as the measure $C_M^{fssd}$ presented in Huang et al. (2021) (FSSD). We recall that FSSD is a OoD-specific method as it uses a *validation* OoD dataset $D_{ood,val}$ in order to learn a linear regressor with both positive and negative samples. In Huang et al. (2021), the authors choose $D_{ood,val} \sim \mathcal{D}_{ood}$, effectively training a binary classifier between $\mathcal{D}_{iod}$ and $\mathcal{D}_{ood}$. For a fair comparison between methods, we perform several experiments using different validation OoD datasets to perform FSSD calibration. Finally, since the output of the classifier can also constitute a valid confidence measure, we also compare our results with the maximum class probability $C_M^{MCP}(x) = \max\big(softmax\big(M(x)\big)\big)$ and the energy-based

(EB) method of Liu et al. (2020) $C_M^{EB}(x) = log\big(\sum_{i=1}^{N} e^{M_{[i]}(x)}\big)$.

| $D_{ood}$ | Metrics | OoD-specific FSSD ($D_{ood,val}$) | | | OoD-agnostic MCP | EB | fNRD | *vanilla* PARTICUL | | *class-based* PARTICUL | |
|---|---|---|---|---|---|---|---|---|---|---|---|
| | | CT | CB | SC | MCP | EB | fNRD | $P=4$ | $P=6$ | $P=4$ | $P=6$ |
| | | | | | $D_{iod}$ = Caltech 101 (heterogeneous) | | | | | | |
| CB | AUROC↑ | / | (95.0) | 28.6 | **87.2** | 83.0 | 46.8 | 72.5 ± 4.8 | 71.7 ± 1.4 | 72.7 ± 0.8 | 69.7 ± 1.7 |
| | AUPR↑ | / | (88.5) | 12.1 | **77.5** | 70.5 | 17.7 | 44.0 ± 8.1 | 47.3 ± 4.6 | 38.6 ± 2.3 | 33.9 ± 2.4 |
| | FPR80↓ | / | (04.5) | 94.5 | **22.1** | 34.1 | 86.3 | 49.9 ± 7.6 | 57.3 ± 1.8 | 49.5 ± 1.1 | 54.6 ± 2.2 |
| SC | AUROC↑ | / | 41.3 | (78.0) | **85.3** | 84.0 | 70.1 | 47.3 ± 10.5 | 41.0 ± 2.4 | 67.1 ± 0.6 | 67.0 ± 2.8 |
| | AUPR↑ | / | 15.3 | (49.2) | **71.8** | 67.1 | 44.8 | 19.9 ± 8.6 | 16.6 ± 3.5 | 24.0 ± 1.0 | 24.4 ± 2.3 |
| | FPR80↓ | / | 96.4 | (45.2) | **26.9** | 32.5 | 68.1 | 88.9 ± 6.0 | 95.5 ± 1.5 | 58.1 ± 0.8 | 60.5 ± 4.2 |
| | | | | | $D_{iod}$ = CUB200 (homogeneous) | | | | | | |
| CT | AUROC↑ | (94.7) | / | 94.5 | **96.1** | 94.0 | 72.1 | 91.8 ± 3.3 | 93.7 ± 0.4 | 93.6 ± 0.3 | 93.1 ± 2.0 |
| | AUPR↑ | (98.3) | / | 98.3 | **99.1** | 98.2 | 91.5 | 97.5 ± 0.8 | 97.7 ± 0.2 | 98.3 ± 0.1 | 98.1 ± 0.6 |
| | FPR80↓ | (7.0) | / | 6.4 | **5.1** | 8.5 | 52.8 | 12.9 ± 5.7 | 9.5 ± 1.3 | 9.4 ± 1.6 | 12.8 ± 5.0 |
| SC | AUROC↑ | 98.6 | / | (99.3) | 98.8 | 97.1 | 88.5 | 97.9 ± 1.8 | **99.1** ±0.1 | 94.4 ± 3.9 | 97.5 ± 1.6 |
| | AUPR↑ | 98.6 | / | (98.9) | **98.6** | 96.9 | 88.8 | 97.3 ± 1.9 | **98.6** ±0.1 | 93.5 ± 4.6 | 96.0 ± 2.0 |
| | FPR80↓ | 0.4 | / | (0.5) | **0.4** | 1.5 | 18.0 | 1.9 ± 1.5 | 0.9 ± 0.1 | 7.9 ± 8.5 | 4.9 ± 4.6 |
| | | | | | $D_{iod}$ = Stanford Cars (homogeneous) | | | | | | |
| CT | AUROC↑ | (99.9) | 99.1 | / | 96.7 | **98.9** | 45.5 | 97.4 ± 1.9 | 98.4 ± 0.6 | 86.8 ± 1.6 | 94.0 ± 3.3 |
| | AUPR↑ | (100) | 99.8 | / | 99.4 | **99.7** | 83.1 | 99.5 ± 0.4 | **99.7** ±0.1 | 97.5 ± 0.4 | 98.9 ± 0.7 |
| | FPR80↓ | (0.1) | 0.8 | / | 1.9 | **0.6** | 79.0 | 3.9 ± 2.9 | 1.8 ± 0.7 | 21.0 ± 6.8 | 8.0 ± 5.5 |
| CB | AUROC↑ | 100 | (99.9) | / | 97.6 | 99.6 | 51.8 | 99.6 ± 0.5 | **99.8** ±0.2 | 95.4 ± 0.8 | 97.0 ± 0.9 |
| | AUPR↑ | 100 | (99.9) | / | 98.0 | 99.7 | 56.6 | 99.7 ± 0.3 | **99.9** ±0.1 | 96.7 ± 0.7 | 97.9 ± 0.7 |
| | FPR80↓ | 0 | (0) | / | 3.7 | 0.4 | 72.5 | 0.1 ± 0.1 | **0.0** ±0.1 | 7.2 ± 2.9 | 2.7 ± 0.9 |

Table 1: AUROC↑, AUPR↑ and FPR80↓ scores (0-100 scale) for different pairs ($D_{iod}, D_{ood}$). SC denotes StanfordCars, CB denotes CUB200 and CT denotes Caltech101. The thick vertical line separates FSSD (OoD-specific) from all other agnostic methods. For FSSD scores, we indicate the validation dataset $D_{ood,val}$ used during the calibration, and scores in parenthesis correspond to the optimal case $D_{ood,val} \sim \mathcal{D}_{ood}$. For OoD-agnostic approaches, we indicate in **bold** the best performing measure for each experiment and each metric.

**Cross-dataset out-of-distribution** In this section, we evaluate the ability of various confidence measures to distinguish the dataset $D_{iod}$ from another dataset $D_{ood} \neq D_{iod}$. As shown in Table 1 and illustrated in Fig. 2, the ability to discriminate $D_{iod}$ from $D_{ood}$ widely differs, depending on the choice of measure and the datasets themselves. In particular, when used as reference distributions, the homogeneous datasets CUB200 and Stanford Cars are in general easier to discriminate than the more heterogenous dataset Caltech-101, across all measures except $C_M^{fnrd}$. This is consistent with the intuition that Caltech101 distribution covers a broader area of the latent space and is therefore harder to circumscribe. As expected, when using $D_{ood,val} \sim \mathcal{D}_{ood}$, $C_M^{fssd}$ becomes a binary classifier with fully-supervised training and displays a high-level of discrimination. However, when using $D_{ood,val} \not\sim \mathcal{D}_{ood}$, the ability of this confidence measure to generalize to unseen datasets is more questionable. Indeed, for Caltech101 v. CUB200 and Caltech101 v. Stanford Cars, we obtain results which vary markedly depending on the choice of validation dataset. Surprisingly, the confidence measure $C_M^{MCP}$ shows good discriminative properties for most pairs ($D_{iod}, D_{ood}$). Without contradicting the conclusions of Hein et al. (2019), this indicates that - outside an adversarial

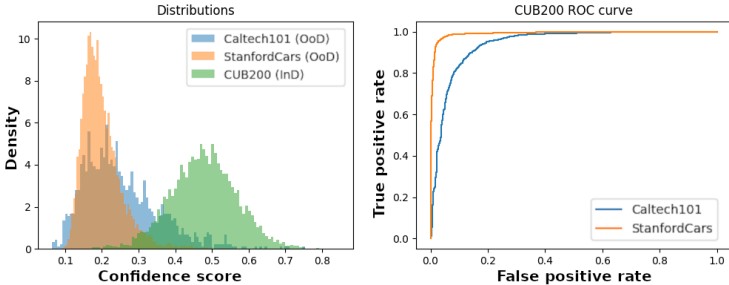

Figure 2: Example of distributions of confidence scores (left) and corresponding ROC curves (right) for a *vanilla* PARTICUL model (6 detectors) trained on CUB200 and tested against Caltech101 and Stanford Cars. Best viewed in color.

setting - using the normalized output logits of a classifier is often enough to distinguish OoD samples. Moreover, we notice that, except for $\mathcal{D}_{iod}$ =StanfordCars, the energy-based method $C_M^{EB}$ does not exhibit better detection performance than the $C_M^{MCP}$ baseline. This experiment also reveals that - in general - the measure $C_M^{fnrd}$ is a poor discriminator of OoD datasets. Both these results may be explained by our choice of datasets which differs from the experiments conducted in these papers, and also emphasize the sensitivity of OoD detection methods to the experimental setting. Results of our PARTICUL-based confidence measure are contrasted and also depend on the choice of architecture (*vanilla* or *class-based*) with respect to the IoD dataset. For CUB200 and Stanford Cars as reference sets, not only does a *class-based* PARTICUL architecture constitute a good OoD detector (with AUROC scores greater than 90% in 67% of the cases), but *vanilla* PARTICUL fully exploits the homogeneity of the IoD dataset to train robust detectors, with AUROC scores greater than 90% in all cases, using fewer detectors compared to the class-based approach. For Caltech101 as the reference set, *class-based* PARTICUL provides similar (CUB200) or better (Stanford Cars) results than *vanilla* PARTICUL, but at the cost of a greater number of detectors, and with an AUROC score lower than $C_M^{MCP}$. This is probably due to the fact that Caltech101 already contains images of birds and cars with similar size, making the distinction between distributions harder.

This first experiment confirms that our PARTICUL-based confidence measure is, in general, suited to perform OoD detection in a cross-dataset setting. For homogeneous training datasets, *vanilla*-PARTICUL shows detection capabilities on par or surpassing OoD-agnostic and OoD-specific (FSSD) approaches. For heterogeneous training datasets, our *class-based* PARTICUL approach improves upon *vanilla*-PARTICUL by learning more specific detectors. Finally, it is interesting to point out that our results are stable, with a standard deviation of the AUROC/AUPR/FPR80 scores of less than 10% in the vast majority of our experiments, and that the number of detectors (4 or 6) does not have a significant impact on the results.

| Perturbation $P$ | Description | Values $\Lambda$ | Sign |
|---|---|---|---|
| Blur | Gaussian blur with kernel $3 \times 3$, standard deviation $\lambda$ | $\lambda=$ 0.0 (no blur), 0.1, 0.3, 0.5, 0.7, 1.0 2.0, 5.0, 10.0 | - |
| Noise | Gaussian noise with ratio $\lambda$ | $\lambda=$ 0.0 (no noise), 0.1, 0.2, 0.3, 0.4, 0.5 0.7, 1.0 | - |
| Brightness | Blend black image with ratio $1 - \lambda$ | $\lambda=$ 0.1, 0.3, 0.5, 0.6, 0.7, 0.8, 0.9, 1.0 (no change) | + |
| Rotation forth | Rotation with degree $\lambda$ | $\lambda=$ 0 (no rotation), 10, 20, 30, 40, 50, 70, 90, 110, 130, 150, 180 | - |
| Rotation back | Rotation with degree $\lambda$ | $\lambda=$ 180, 210, 240, 270, 300, 320, 340, 350, 360 (no rotation) | + |

Table 2: Description of the perturbations used in this paper. For each perturbation, we define the variable parameter $\lambda$, its range of values $\Lambda$ and the expected sign of the correlation. '+' (resp. '-') indicates that the confidence measure is expected to increase (resp. decrease) with the intensity of the perturbation. For gaussian noise, we dynamically compute a standard deviation for each input $x$ as $\sigma(x) = \lambda \times \big( \max(x) - \min(x) \big)$ (similar to the method used in Smilkov et al. (2017)).

| Arch. | $\mathcal{D}_{ind}$ | Pert. $P$ | MCP | EB | fNRD | vanilla PARTICUL | | class-based PARTICUL | |
|---|---|---|---|---|---|---|---|---|---|
| | | | | | | $P=4$ | $P=6$ | $P=4$ | $P=6$ |
| Resnet50 | CT | Blur (-) | -1.00 | -0.85 | 1.00 | -0.91 ± 0.06 | -0.95 ± 0.00 | -0.95 ± 0.00 | -0.95 ± 0.00 |
| | CB | | -0.90 | 0.61 | 1.00 | 0.95 ± 0.00 | 0.95 ± 0.00 | 0.95 ± 0.00 | 1.00 ± 0.00 |
| | SC | | -1.00 | -0.95 | 1.00 | -0.25 ± 0.60 | 0.50 ± 0.14 | -0.85 ± 0.00 | -0.86 ± 0.03 |
| | CT | Noise (-) | 0.05 | 0.5 | -1.00 | -1.00 ± 0.00 | -1.00 ± 0.00 | -1.00 ± 0.00 | -1.00 ± 0.00 |
| | CB | | -0.24 | 0.12 | -1.00 | -0.69 ± 0.00 | -0.68 ± 0.08 | -0.78 ± 0.10 | -0.67 ± 0.08 |
| | SC | | -0.29 | -0.29 | -1.00 | -0.77 ± 0.03 | -0.82 ± 0.01 | -0.61 ± 0.61 | -0.93 ± 0.12 |
| | CT | Brightness (+) | 0.98 | 1.00 | -1.00 | 0.97 ± 0.04 | 0.98 ± 0.04 | 1.00 ± 0.00 | 1.00 ± 0.00 |
| | CB | | 1.00 | 1.00 | -0.31 | -0.01 ± 0.52 | 0.23 ± 0.11 | 0.78 ± 0.20 | 0.67 ± 0.20 |
| | SC | | 1.00 | 1.00 | -1.00 | 0.98 ± 0.03 | 0.90 ± 0.18 | 1.00 ± 0.00 | 1.00 ± 0.00 |
| | CT | Rotation forth (-) | -0.30 | -0.28 | -0.24 | -0.41 ± 0.18 | -0.42 ± 0.05 | -0.51 ± 0.08 | -0.52 ± 0.07 |
| | CB | | -0.95 | -0.89 | -0.42 | -0.82 ± 0.20 | -0.94 ± 0.01 | -0.94 ± 0.01 | -0.93 ± 0.00 |
| | SC | | -0.76 | -0.76 | -0.56 | -0.71 ± 0.06 | -0.67 ± 0.09 | -0.86 ± 0.08 | -0.89 ± 0.08 |
| | CT | Rotation back (+) | 0.52 | 0.65 | 0.20 | 0.51 ± 0.09 | 0.44 ± 0.09 | 0.68 ± 0.10 | 0.73 ± 0.06 |
| | CB | | 0.95 | 0.90 | 0.32 | 0.77 ± 0.32 | 0.94 ± 0.01 | 0.98 ± 0.03 | 0.98 ± 0.03 |
| | SC | | 0.72 | 0.72 | 0.32 | 0.78 ± 0.06 | 0.74 ± 0.03 | 0.84 ± 0.05 | 0.89 ± 0.07 |
| VGG19BN | CB | Blur (-) | -0.86 | -1.00 | 1.00 | -1.00 ± 0.00 | -1.00 ± 0.00 | -1.00 ± 0.06 | -0.96 ± 0.06 |
| | | Noise (-) | -1.00 | -1.00 | -1.00 | -1.00 ± 0.00 | -1.00 ± 0.00 | -0.92 ± 0.06 | -0.97 ± 0.05 |
| | | Brightness (+) | 0.98 | 1.00 | -0.98 | 1.00 ± 0.00 | 1.00 ± 0.00 | 1.00 ± 0.00 | 1.00 ± 0.00 |
| | | Rotation forth (-) | -0.95 | -0.94 | -0.42 | -0.95 ± 0.00 | -0.95 ± 0.00 | -0.95 ± 0.00 | -0.95 ± 0.00 |
| | | Rotation back (+) | 0.95 | 0.93 | 0.35 | 0.94 ± 0.01 | 0.92 ± 0.01 | 0.92 ± 0.01 | 0.92 ± 0.01 |

Table 3: Spearman rank correlation coefficient between the intensity $\lambda$ of a perturbation applied to dataset $\mathcal{D}$ and the average confidence measure $\Gamma(C_M, \mathcal{D}, P, \lambda)$ computed over the dataset. For each perturbation $P$, the expected correlation sign is recalled in parenthesis. In red, we highlight experiments where the distributions are either uncorrelated (small correlation coefficient), inversely correlated (opposite coefficient sign) or unstable (high standard deviation for PARTICUL-based measures). For CUB200, we also provide the results of the experiments performed using a VGG19 classifier (with batch normalization). Best viewed in color.

**Sensitivity to dataset perturbation** In this experiment, we study the behaviour of confidence measures using the various perturbations described in Table 2. Due to its instability during the previous experiment and its OoD-specific setting, $C_M^{fssd}$ is not considered here. As shown in Table 3, the four confidence measures under test exhibit very different behaviors depending on the nature of the perturbation, but also sometimes depending on the reference dataset. Indeed, although $C_M^{MCP}$ and $C_M^{EB}$ seem in general sensitive to blur and brightness perturbations, they are not correlated with gaussian noise, opening the door, if it were employed, to adversarial attacks (Szegedy et al. (2014); Hein et al. (2019)). Conversely, while $C_M^{fnrd}$ is strongly sensitive to noise, any perturbation reducing the amplitude of neuron activation values (blur, brightness) has the opposite effect of increasing this confidence measure. In general, both $C_M^{vP}$ and $C_M^{cP}$ correlate with each type perturbation. However, we notice that results on CUB200 (and Stanford Cars, in a lesser extent) for blur and brightness suffer the same issue as $C_M^{fnrd}$ (inverse correlation compared to the correct correlation) when using our baseline classifier (ResNet50). Finally, rotations seem harder to detect, with MCP, EB and PARTICUL-based measures achieving similar results, while $C_M^{fnrd}$ is only weakly correlated to the rotation angle. Interestingly, when using a VGG19BN (Simonyan & Zisserman (2015)) classifier trained on CUB200, we obtain better results for $C_M^{MCP}$/$C_M^{EB}$ (noise sensitivity) and our PARTICUL-based approach (blur, brightness), while confirming the drawbacks of $C_M^{fnrd}$ (blur, brightness). This indicates that not only does the robustness of a given measure depends on the type of perturbation and on the IoD dataset, but it also seems to rely on the architecture of the classifier.

**A step towards more explainable confidence measure** One of the key advantages of the PARTICUL-based confidence measure resides in the possibility for a human to visualize the detected patterns during the inference of the classifier in order to trace back the contribution of each detector. As in Xu-Darme et al. (2022), we use the Smoothgrads (Smilkov et al. (2017)) algorithm to highlight the most salient area of the image for each detector. As illustrated in Fig. 3a, images from $D_{iod}$ can be used to extract a visual reference of the pattern highlighted by each detector. When processing a new image through the classifier $M$, patterns can be compared to these visual references to better understand the confidence score. For example, some detectors trained on the StanfordCars dataset seem to be able to recognize similar patterns on an image of a motorcycle (row 3), which is to be expected from a visual point of view. Secondly, visualization can also help illustrate the behaviour of each detector *w.r.t.* a perturbation of the image, as illustrated in Fig. 3b and Fig. 3c. In particular, it is interesting to notice that contrary to our initial intuition, our PARTICUL-based confidence measure does not steadily decrease for angles ranging from 0° to 180°, but rather displays some invariance to

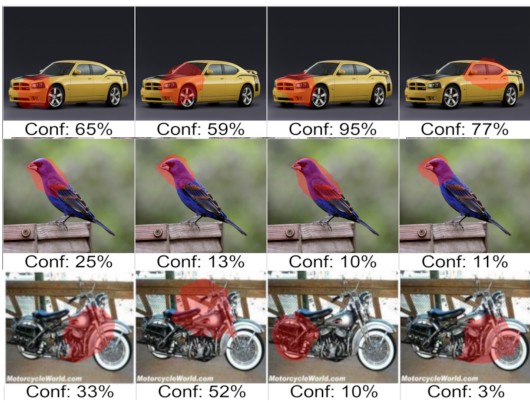



(a) Pattern visualization and confidence scores of 4 detectors (one per column) trained using *vanilla* PARTICUL on StanfordCars and applied to images from StanfordCars (row 1), CUB200 (row 2) and Caltech101 (row 3). The first image can be used as a visual reference.

(b) Impact of increasing gaussian noise ratio on four detectors trained with *class-based* PARTICUL on Caltech101 (*motorbikes* class. Averaging the confidence scores across all detectors creates an ensemble effect for more robustness.).

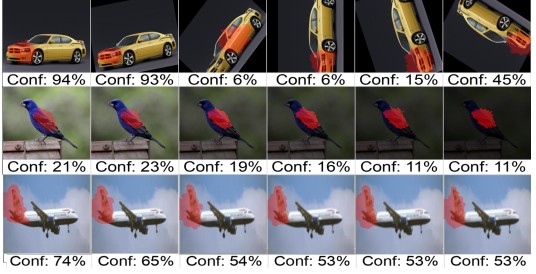

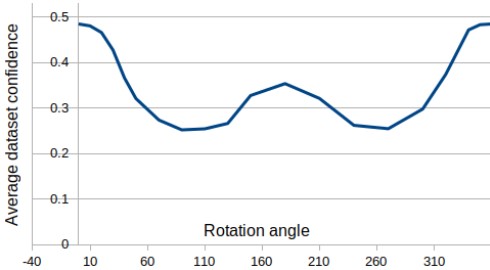

(c) Evolution of the confidence score of three detectors trained on StanfordCars (row 1), CUB200 (row 2) and Caltech101 (row 3) respectively *w.r.t.* to rotation (row 1), changes in brightness (row 2) and gaussian blur (row 3).

(d) Evolution of the average confidence of a *vanilla* PARTICUL detector across the Stanford Cars dataset for various rotation angles. Rather than the expected V-shape curve, we observe a local maxima for 180° rotation (see (c), row 1).

Figure 3: Using pattern visualization to better understand our PARTICUL-based confidence measure.

horizontal symmetry (Fig.3d). Finally, as shown in Fig. 3b, although each of our detectors might not be strongly sensitive to a given perturbation, averaging scores across all detectors partially mitigates the variations of individual detectors and creates an ensemble effect that improves the overall stability of the metric, without the cost of training several classifiers (Nguyen et al. (2020)).

## 6 CONCLUSION & FUTURE WORKS

In this paper, we have demonstrated how the detection of recurring patterns, using the PARTICUL algorithm, can be exploited to develop confidence measures with the ability to discriminate between IoD and OoD inputs, while enabling a form of visualisation of the detected patterns. Our experiments showed that it offers mostly consistent results in the context of two different OoD detection benchmarks. In particular, we introduced a new OoD benchmark based on perturbations of the reference dataset with increasing magnitude which offers a quantifiable evaluation of the discrepancy between IoD and OoD inputs. We also showed that the robustness of all tested metrics can depend on the architecture of the underlying classifier and the processed dataset. We believe that experiments with multiple architectures should be systematically conducted in future proposals for measuring Out-of-Distribution. Moreover, since the datasets we used were not strictly exclusive (Caltech101 contains images of both birds and cars), a proper ablation study (retraining a classifier without these classes) should be performed to verify whether the separability of distributions actually improves. Finally, we also wish to extend our benchmark to a more large scale dataset such as ImageNet Deng et al. (2009) and to study the possibility of applying our PARTICUL-based confidence measure to other architectures such as vision transformers Kolesnikov et al. (2021).

**Reproducibility**    All the code necessary to reproduce our experimental results is available at the following link: https://anonymous.4open.science/r/iclr23_submission_596/

| Dataset | Architecture | Image size | Accuracy | # epochs | Optimizer | | |
|---|---|---|---|---|---|---|---|
| | | | | | alg | learn. rate | decay/mom. |
| Caltech 101 | | 224 | 86.8% | 30 | Adam | $10^{-3}$ | / |
| CIFAR 100 | Resnet50 | | 71.3% | 30 | Adam | $10^{-3}$ | / |
| CUB200 | | 448 | 83.6% | 80 | SGD | $10^{-3}$ | $10^{-4}$ / 0.9 |
| StanfordCars | | | 90.5% | 80 | SGD | $10^{-3}$ | $10^{-4}$ / 0.9 |
| CUB200 | VGG19BN | 448 | 83.0% | 100 | SGD | $10^{-3}$ | $10^{-4}$ / 0.9 |

Table 4: Baseline classifiers training parameters.

Table 4 summarizes the parameters used when training our baseline classifiers. Note that for CUB200 and Stanford Cars, we perform data augmentation through a random cropping of size $448 \times 448$ on the training images (for CIFAR100 and Caltech101, no data augmentation is used) and reduce the learning rate by a factor 10 every 30 epochs.

For reproducibility purposes, we performed 3 random initializations of the PARTICUL detectors for both cross-dataset and dataset perturbation OoD settings. We are providing averaged scores along with corresponding unbiased standard deviation. In our tables, average score is on the left of the $\pm$ symbol, standard deviation is on the right.

When training our PARTICUL detectors (either in the *vanilla* or *class-based* versions), we use the RMSprop optimizer, with a learning rate of $10^{-4}$ and a decay of $10^{-5}$, for 30 epochs. We do not use any data augmentation except resizing each image to the appropriate size. In all experiments, we try to maximize the diversity of our detectors by setting the ratio $\lambda$ of the Unicity constraint to 1 (see Xu-Darme et al. (2022)). The training process of the detector is in itself very fast: as an indication, when training 6 *vanilla*-Particul detectors on the CUB200 dataset (images of size $448 \times 448$), we are able to process around 1000 images per min on a Quadro T2000 mobile.

For the implementation of FSSD, we used the code available on GitHub (https://github.com/megvii-research/FSSD_OoD_Detection), with the following modifications:

- Allow the choice of a validation set $D_{ood,val} \not\sim D_{ood}$ for a fairer comparison betwen methods
- Use the entire test sets during the evaluation rather than subsampling $D_{ood}$ to match the size of $D_{iod}$ (would improve AUPR score in some cases).

Note that during training, we did not perform hyperparameter tuning due to its computational cost.

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
