# OpenReview forum: "Interpretable Out-of-Distribution Detection using Pattern Identification"
_ICLR.cc/2023/Conference — Submitted to ICLR 2023_

### Official Review · Reviewer_DuyP · 2022-10-24

**Confidence:** 3
**Correctness:** 3
**Technical Novelty And Significance:** 2
**Empirical Novelty And Significance:** 2
**Recommendation:** 3

**Clarity, Quality, Novelty And Reproducibility:**

The paper is clearly written overall. However, it substantially relies on the previously proposed PARTICUL algorithm and the novel finding seems marginal. Their experimental results seem clearly reproducible as the detailed description of the parameters and implementation settings were provided.



**Strength And Weaknesses:**

[+] The problem statement and the proposed approach are clear and straightforward. The proposed method provides a graphical interpretation of OoD by mining patterns in the latent representations in CNN-based classifiers. And the key aspects of this work, the robustness and interpretability, have been investigated through experiments.

[-] The major concern is the proposed approach's contribution and novelty. A considerable part of the proposed model appears to overlap with the previous model  (PARTICUL, Xu-Darme et al.(2022)). They modified the PARTICUL algorithm to be a more interpretable and robust confidence measure for OOD detection, but it does not show substantial improvement or novelty in the methodological aspects.

[-] The paper mainly investigates the characteristics of their PARTICUL-based approach itself. It needs to be compared with the standard benchmark for OoD (e.g., Open-OoD).


**Summary Of The Paper:**

The authors showed how the identification of recurrent patterns from the latent space of a CNN can be utilized for OoD detection problems. They utilize the existing PARTICUL algorithm to distinguish the anomaly patterns from the input and evaluate their approach in two modalities (cross-dataset OoD and perturbation OoD).

**Summary Of The Review:**

While the proposed approach is well described and the experiment results somewhat show the robustness and interoperability of the proposed model, its contribution and significance appear marginal because it is largely based on the previous work. Experimental validation could also be improved by adding other widely used benchmarks for OOD.

---

> ### Author Response · Authors · 2022-11-18
> **Response to reviewer DuyP**
>
> We agree that, rather than presenting our modifications to the Particul algorithm as the main subject of our paper, the novelty aspect of our work largely resides in the fact that:
> 1) We showed that it is possible to employ existing work from the field of explainable AI to build more interpretable OoD detectors that exhibit an accuracy on par with traditional OoD techniques.
> 2) In addition to the evaluation of these OoD measures in a cross-dataset setting, we propose a new benchmark for OoD detection based on perturbations of the reference dataset with transformations of increasing intensity. This setting provides a quantifiable measure (perturbation intensity) of the difference between the IoD and OoD dataset which can act as a reference value and a basis of comparison between OoD measures. In addition to traditional benchmarks such as OoD detection or anomaly detection, we believe that such setting very provide valuable information regarding the sensitivity of OoD detectors. In the context of critical applications, this method may be extended to more realistic transformations simulating real-world changes in order to check the sensitivity of the detectors in a controlled environment.
> We thank the reviewer for this comment and we have modified the phrasing of our contribution accordingly.
>
> We would have liked to perform an in-depth comparison of our approach with other methods using the Open-OoD benchmark, at least using ImageNet as a reference dataset (see our response to Reviewer CPyj regarding low resolution datasets). Unfortunately, we do not currently have the computing capability to calibrate our detectors on such a large dataset but we plan to do it as future work.

---

### Official Review · Reviewer_CPyj · 2022-10-24

**Confidence:** 4
**Correctness:** 3
**Technical Novelty And Significance:** 2
**Empirical Novelty And Significance:** Not applicable
**Recommendation:** 3

**Clarity, Quality, Novelty And Reproducibility:**

In my view, this paper is an extension of a local-based confidence detection method for OOD detection tasks. The novelty is limited.

Some experiments are interesting but it is not well demonstrated, like the experiments in Table 3.

The paper leaks comparisons with current SOTA methods on the common-used datasets on OOD detection tasks, making the paper less supported.

**Strength And Weaknesses:**

Strength:

Extend the local-based confidence detection method to OOD detection task and further explore different perturbations which may contribute to the OOD detection.

Weakness:

Current work in Image OOD detection methods usually uses MINIST, ImageNet, and Cifar100 as the validation dataset, while the authors choose Caltech-101, CUB-200, and StanfordCars as the validation dataset. Is there any special reason for such a dataset choice?
In addition, the chosen datasets are focused on quite different object categories, (CUB -- birds, and StanfordCars -- cars), which can make the OOD images easier to be distinguished at the feature level. I suggest the authors also complete the experiments on the current popular OOD detection datasets so and compare the performance with proposed methods, i.e. ODIN, Maha, and FSSD.

The Table 3. Is not very clear. In the table, the authors show correlation coefficients between lambda and confidence measures, while do not provide the quantitative result for each test case. It is not clear the quantitative performance of each perturbation case on different datasets. And more analysis on the inversely correlated cases. For example, why do the blur cases and MCP have a positive correlation, while having an inverse correlation with fNRD?


**Summary Of The Paper:**

This paper extends a local-based confidence detection method to the image OOD detection task. They further examine the proposed method with variant kinds of perturbations, i.e. blurring, noise, brightness, and rotation. The model is tested on Caltech-101, CUB-200, and StanfordCars.

**Summary Of The Review:**

In general, I think the novelty of this paper is limited and still has some shortcomings in the experiments. Therefore, I do not recommend this paper generally.

---

> ### Author Response · Authors · 2022-11-18
> **Response to reviewer CPyj**
>
> The results of our experiments on the CIFAR100 dataset are present in the supplementary material but could not be integrated properly in the main paper due to lack of space. Such experiments mostly confirmed what has been shown for Caltech101 in the main paper. We agree with the reviewer that comparison between methods using common datasets is informative. However, although low resolution datasets such as MNIST and CIFAR100 are indeed used in most related works, we believe that such datasets are not necessarily representative of real-life computer vision problems and that we should refrain from using it, lest we propose methods that do not scale to more realistic settings. We have indeed not performed any experiments on the ImageNet dataset due to limited computing resources but plan to do it as a future work. Although our choice of datasets (2 fine-grained, 2 heterogeneous - including CIFAR100) might be debatable in a cross-dataset OoD setting, we argue that in a perturbation OoD setting, the difference in nature of these datasets vanishes because we are trying to distinguish a given image from its perturbed counterpart.
>
> As mentioned in our response to Reviewer 2viJ, ODIN and Maha are not included in our experiments because both methods are similar to FSSD in the sense that they require the use of a validation OoD dataset and are therefore not directly comparable to our approach.
>
> During our perturbation experiments described in Table 3, we use Spearman rank correlation coefficients rather than alternative quantitative results (e.g. the derivative of the measure with respect to the intensity) for the following reasons:
> 1) Averaging the derivative across multiple intensity values might hide the fact that there is no correlation between the measure and the perturbation (e.g. MCP/EB versus gaussian noise)
> 2) Having a measure M1 dropping more rapidly than another measure M2 when increasing the intensity of a perturbation would not necessarily imply that M1 is better than M2, especially when both measures have different calibration methods.
> Such reasons have been stated more clearly in the paper and we thank the reviewer for pointing this to us.
>
> Finally, regarding the results of fNRD in a blur and brightness setting, we believe - as stated in the paper -  that fNRD shows an increasing confidence because such operations tend to reduce the amplitude of the activation of the neurons that are monitored by this measure, therefore reducing their probability to activate outside of their typical range.

---

### Official Review · Reviewer_2viJ · 2022-10-26

**Confidence:** 4
**Correctness:** 2
**Technical Novelty And Significance:** 2
**Empirical Novelty And Significance:** 2
**Recommendation:** 3

**Clarity, Quality, Novelty And Reproducibility:**

The quality of writing should be improved, and the baseline setting should be changed in the experiment. Also, although the interpretable property is attractive, I think the OoD detection performance of the proposed method is not sufficient.

**Strength And Weaknesses:**

Strength
1. The interpretability of the proposed method allows us to understand the cues of the data detected as anomalies.


Weakness
1. I believe that the writing quality of the manuscript should be improved.
- The format of the paper is different from the ICLR format. It is necessary to fix the space between each paragraph and the space at the bottom of the page.
- There is no explanation of how PARTICUL is trained. In particular, on page 4, there is no explanation of how L_l and L_u are calculated.
- In Figure 3, each image seems to be unaligned, and in (b), the noise suddenly appears on both sides of the image.

2. The OoD detection methods used as the baseline should be changed. Among those models [1,2,3,4,5] that are generally used as baselines in the OoD detection community, only [1] is used in this paper. If the authors want to use fNRD and FSSD rather than using these models as baselines, I think they should provide a clear reason for it.

3. I have concerns about the applicability and practicality of the proposed method. First of all, the proposed model seems to be applicable only to those models using the CNN architecture. Therefore, unlike the existing approaches [1,2,3,4,5], it seems impossible to utilize the proposed method for other architecture such as Transformer or RNN. Also, I think training p detectors for each class requires a non-trivial amount of time.

4. The improvement of the OoD detection performance is marginal. In Table 1, the proposed method does not consistently perform better than MCP, which is a highly simple method. Also, FPR95 is generally used as an evaluation metric rather than FPR80 in the community. I wonder if there is a particular reason for the authors to use the FPR80.

[1] A Baseline for Detecting Misclassified and Out-of-Distribution Examples in Neural Networks, Hendrycks et al., ICLR 2017

[2] Enhancing The Reliability of Out-of-distribution Image Detection in Neural Networks, Liang et al., ICLR 2018

[3] A Simple Unified Framework for Detecting Out-of-Distribution Samples and Adversarial Attacks, Lee et al., NeurIPS 2018

[4] Deep Anomaly Detection with Outlier Exposure, Hendrycks et al., ICLR 2019

[5] Energy-based Out-of-distribution Detection, Liu et al., NeurIPS 2020


**Summary Of The Paper:**

In this paper, the authors propose a new out-of-distribution (OoD) detection method. Specifically, the PARTICUL algorithm is used to find recurring patterns in the training dataset. Then, the degree to which each pattern is included in the data is measured and used as an OoD score. Also, thanks to the use of PARTICUL, the results of the proposed OoD detection method are visually interpretable. Finally, the authors conducted an experiment to detect 1) OoD data of a completely different class from the IoD dataset and 2) OoD data with a distributional shift.

**Summary Of The Review:**

I believe the quality of the writing, the experimental settings, and the quantitative results should be improved.

---

> ### Author Response · Authors · 2022-11-18
> **Response to reviewer 2viJ**
>
> As mentioned by the reviewer, methods such as ODIN [2], Maha [3], OE [4]  and Energy score [5] are often used as baselines for comparing OoD methods and have been added to our related work. However, we argue that:
> 1) Both ODIN [2] and Maha [3] require the use of an OoD validation set for calibrating the measure. As shown in the paper, our approach does not require such a validation set and is “OoD-agnostic”. Moreover, we have shown with experiments on FSSD that the results of such approaches may vary depending on the choice of validation set. This reliance on the validation set is also incidentally mentioned in OE [4]: "Unlike Liang et al. (2018); Lee et al. (2018), in this work we train our method without tuning parameters to fit specific types of anomaly test distributions, so our results are not directly comparable with their results."
> 2) OE requires to fine-tune the network to satisfy the L_OE objective along with the traditional classification objective, a process that is not needed in our approach and may not be applicable in an industrial context where the model might somewhat be treated as a black box (allowing us only to probe some layers, but not change the existing weights).
> 3) We thank the reviewer for suggesting the integration of the Energy score [5], which is directly comparable to our approach, into our experiments. We took the time to implement, test this approach in our experiments and integrate the results in our paper.
>
> Regarding the applicability of our method to Transformers and RNNs, we have indeed not performed any experiments with such architectures and this constitutes a possible and interesting future work.
>
> Regarding the method for training the PARTICUL detectors, we use the loss functions (L_l and L_u) and learning parameters indicated in Xu-Darme et al (2022). For homogeneous datasets (i.e. fine-grained datasets), we directly use the unsupervised algorithm described in this paper. For more heterogeneous datasets, the learning process is applied on a per-class basis, using again the same loss functions. The difference is that the choice of detectors to select during inference is dependent on the model's predicted class rather than the ground truth, making the process dependent on the model accuracy. The training process in itself is not time-consuming due to the fact that the classifier network is frozen and that each Particul detector is a simple 1x1xD convolutional kernel (e.g. for CIFAR100 or Caltech101, this represent only a few hundred kernels). Moreover, even in a class-based Particul setting, only the detectors of the target class are updated during backpropagation. As an example, on a simple Quadro T2000 Mobile, we are able to process approximately a 1000 images/min during training. We thank the reviewer for mentioning this lack of clarity and we have updated the paper accordingly.
>
> Regarding the performance of our approach, we do agree that the gain with respect to MCP or EB may appear marginal in a cross-dataset setting: these relatively good results of MCP also surprised us and we felt that it was worth confirming that the use of MCP in environments with limited computing resources (such as embedded systems) is still a viable option. However, our approach does benefit from an interpretable element that is missing from MCP/EB.
>
> Finally, regarding the use of FPR80 rather than FPR95, we initially used this metric in order to be compatible with the results shown in the FSSD paper.

---

> > ### Comment · Reviewer_2viJ · 2022-11-25
> > **Response**
> >
> > I appreciate the authors' responses and newly included experiments. However, I still have concerns about the overall OoD detection performances, including the results of the EB. Also, I believe that the authors should report the results of FPR 95. Therefore, I would like to keep my score.

---

### Official Review · Reviewer_w3oX · 2022-11-11

**Confidence:** 4
**Correctness:** 3
**Technical Novelty And Significance:** 2
**Empirical Novelty And Significance:** 2
**Recommendation:** 3

**Clarity, Quality, Novelty And Reproducibility:**

The overall presentation is ok, however, it lacks insightful discussion about the proposed technique. The novelty and contribution are also somehow limited. The authors provide the code for reproduce the results, which is a good sign for reproducibility.

**Strength And Weaknesses:**

Strength:

- Important problem
- Sound and feasible solution
- Extensive evaluation to demonstrate the potential usefulness
- Promise results

Weakness:

- Limited novelty, which is mostly an incremental extension of previous work
- Limited contribution and insight, the authors mostly discuss how, but unclear why and what is the insight behind the techniques.
- Interpretable results would be highly demanding, however, this paper again only leverages another existing method for visualization.
- Not very clear about the implications of the perturbation magnitude and their relation to OoD.

**Summary Of The Paper:**

This paper proposes a pattern identification-based OoD detection technique and further leverages a visualization-based approach to interpret the obtained results, providing more tangible results from the human’s perspective. The proposed techniques are based on a previous work PARTICUL, and the evaluation is performed to demonstrate the usefulness of the proposed methods.

**Summary Of The Review:**

The paper proposes a sound and feasible method for OoD detection, the evaluation also demonstrates the potential of the proposed method. However, it still posts a few major concerns considering its current status:

- Although the proposed techniques are overall feasible, the novelty is limited, which is mostly based on previous work. The insight of the proposed techniques is also not well discussed.

- The contribution is also limited especially in terms of technical contribution to push the research of OoD, which are mostly based on two previous work for OoD detection, and post an explanation of the results.

- As for the robustness analysis via perturbation, I think this could be interesting. However, the current status does not give too much insight into the impacts and implications of the perturbation, and how the OoD could be connected to robustness, e.g., what is the relation, what is the gap there.

Overall, I do believe the problem this work intends to solve is important. However, it still needs further enhancement to clarify the novelty and contribution.

---

> ### Author Response · Authors · 2022-11-18
> **Response to reviewer w3oX**
>
> Regarding the novelty aspect of our submission, we agree that the objectives of the paper might have been stated differently (see our proposition in our response to Reviewer DuyP).
>
> Regarding our perturbation-based OoD benchmark, the goal is not to evaluate the robustness of the classifier but rather to progressively "move" the OoD dataset (perturbed dataset) away from the original dataset IoD using transformations of increasing intensity and to check whether the OoD measures under test are not only able to detect this drift, but also able to correlate with the intensity of the transformation (see our response to Reviewer DuyP).

---

### Author Response · Authors · 2022-11-18
**Summary of changes following the reviews**

Once again, we wish to thank all the reviewers for their constructive and very helpful comments. We have made some changes to our paper accordingly and we hope that we have addressed most of the concerns that have been raised. Here is a summary of the changes that have been made:
1) The abstract, the introduction and the conclusion have been modified in order to emphasize:
- That the novelty of our approach resides in the use of techniques from XAI to build more interpretable OoD detectors with performance on par with existing methods of the field;
- That we have proposed a new benchmark for OoD based on perturbations that provides a quantifiable measure (perturbation intensity) of the difference between the IoD and OoD dataset, which can act as a reference value and a basis of comparison between OoD measures
2) The related work has been updated with missing references pointed by the reviewers. We have also reorganized this section to better justify which methods (OoD-agnostic) and which datasets have been selected in our experiments.
3) We implemented and tested the Energy-based OoD detection method which is directly comparable to our approach. The results of these experiments have been integrated in the tables of the paper.

---

### Decision · Program_Chairs · 2023-01-20

**Decision:**

Reject

**Justification For Why Not Higher Score:**

NA

**Justification For Why Not Lower Score:**

NA

**Metareview: Summary, Strengths And Weaknesses:**

This paper studies interpretable out-of-distribution detection, which is an interesting and important problem. The interpretability perspective of OOD detection is particularly underexplored. The proposed method fills the gap, and allows understanding and visualizing the cues of the data detected as anomalies. In this regard, the method has advantages and strengths over prior OOD detection methods.

There are several common concerns raised by the reviewers, such as

- (1) The novelty of the method is limited. A considerable part of the proposed model appears to overlap with the previous model (PARTICUL, Xu-Darme et al.(2022)).
- (2) During the rebuttal, the authors added the energy score baseline (Liu et al. 2020), which strengthens the manuscript. However, the paper necessitates a further comparison with state-of-the-art OOD detection methods developed more recently (particularly the ones published in 2022).

Aside from the above concerns, interpretability methods can be fragile and may produce unreliable recurring patterns --- this may negatively impact OOD detection. The paper can be strengthened by investigating the failure cases more thoroughly and providing a complete picture of its capability and incapability.

The paper is rejected since all four reviewers unanimously voted for this decision.